# Human Indoleamine 2,3-dioxygenase 1 (IDO1) Expressed in Plant Cells Induces Kynurenine Production

**DOI:** 10.3390/ijms22105102

**Published:** 2021-05-12

**Authors:** Michele Bellucci, Andrea Pompa, Carine De Marcos Lousa, Eleonora Panfili, Elena Orecchini, Elisa Maricchiolo, Daniele Fraternale, Ciriana Orabona, Francesca De Marchis, Maria Teresa Pallotta

**Affiliations:** 1Institute of Biosciences and Bioresources, National Research Council of Italy, 06128 Perugia, Italy; michele.bellucci@ibbr.cnr.it (M.B.); andrea.pompa@uniurb.it (A.P.); 2Department of Biomolecular Sciences, University Carlo Bo, 61029 Urbino, Italy; e.maricchiolo@campus.uniurb.it (E.M.); daniele.fraternale@uniurb.it (D.F.); 3Centre for Biomedical Sciences, School of Clinical and Applied Sciences, Leeds Beckett University, Leeds LS13HE, UK; C.De-Marcos-Lousa@leedsbeckett.ac.uk; 4Centre for Plant Sciences, Faculty of Biological Sciences, University of Leeds, Leeds LS29JT, UK; 5Department of Medicine and Surgery, University of Perugia, 06128 Perugia, Italy; eleonora.panfili@unipg.it (E.P.); elena.orecchini@studenti.unipg.it (E.O.); ciriana.orabona@unipg.it (C.O.)

**Keywords:** protoplasts, kynurenine, genetic transformation

## Abstract

Genetic engineering of plants has turned out to be an attractive approach to produce various secondary metabolites. Here, we attempted to produce kynurenine, a health-promoting metabolite, in plants of *Nicotiana tabacum* (tobacco) transformed by *Agrobacterium tumefaciens* with the gene, coding for human indoleamine 2,3-dioxygenase 1 (IDO1), an enzyme responsible for the kynurenine production because of tryptophan degradation. The presence of IDO1 gene in transgenic plants was confirmed by PCR, but the protein failed to be detected. To confer higher stability to the heterologous human IDO1 protein and to provide a more sensitive method to detect the protein of interest, we cloned a gene construct coding for IDO1-GFP. Analysis of transiently transfected tobacco protoplasts demonstrated that the IDO1-GFP gene led to the expression of a detectable protein and to the production of kynurenine in the protoplast medium. Interestingly, the intracellular localisation of human IDO1 in plant cells is similar to that found in mammal cells, mainly in cytosol, but in early endosomes as well. To the best of our knowledge, this is the first report on the expression of human IDO1 enzyme capable of secreting kynurenines in plant cells.

## 1. Introduction

The production of recombinant proteins in plants dates back to the end of 1980s [1], and since then many recombinant proteins have been produced in plants for different uses, spanning from biomedical applications to industrial production [2,3]. Plant platforms employing different expression systems have been developed, with the main distinction between stable transformation, where transgenes can be integrated either in the nuclear DNA or in the plastid DNA [4,5], and transient expression [6]. Foreign genes have been transferred into the plants by *Agrobacterium tumefaciens*-mediated transformation methods [7,8] or other delivery systems including particle bombardment, PEG-mediated DNA uptake in wall-free protoplasts [9], and nanomaterial-based delivery devices [10]. However, plant expression procedures share some common characteristics such as the production of favourable post-translational modifications and the absence of contaminating endotoxins and animal viruses, which are serious issues in bacterial and eukaryotic expression systems. Several reports have shown the successful expression of bioactive human proteins in plants [11,12,13].

Indoleamine 2,3-dioxygenase 1 (IDO1) represents one of the most interesting proteins which links an ancient metabolic pathway with immune regulation. IDO1 is a heme-containing enzyme that catalyses the oxidative cleavage of the indole ring in L-tryptophan, regulating the catabolism of this essential amino acid at an initial, rate-limiting level in a specific pathway. This activity leads to the kynurenine production, which is the upstream metabolite along the so-called kynurenine pathway [14]. IDO1 is widely recognised as an authentic immune regulator capable of fine-tuning immune responses not only in pregnancy [15] but also in several pathological conditions, such as autoimmune diseases [16,17], chronic inflammation [18], transplantation [19], and neoplasia [20]. IDO1 immunoregulatory effects are mainly mediated by dendritic cells (DCs), the most potent antigen-presenting cells in the mammal immune system that, upon IDO1 upregulation, acquire tolerogenic functions. In fact, through its enzymatic activity, IDO1 generates both tryptophan deprivation in local tissue microenvironments and the formation of bioactive metabolites collectively called kynurenines. In particular, several kynurenines reinforce the immunoregulatory circuit in mammals by acting as a ligand of aryl hydrocarbon receptor (AhR) in both T cells and DCs [21]. As a result, IDO1-expressing DCs mediate multiple effects on T lymphocytes, including inhibition of proliferation, apoptosis, and differentiation towards a regulatory phenotype [22]. Moreover, apart from being a cytoplasmic enzyme, IDO1 also acts as a signal-transducing molecule by binding tyrosine phosphatases in its immunoreceptor tyrosine-based inhibitory motif (ITIM) domains [23] and anchoring to the early endosomes. This localisation is a result of the IDO1 interaction with activated class IA PI3K subunits [24]. Thus, the IDO1 ability to restrain inflammation depends on two distinct functions of the IDO1 protein: one is enzymatic (tryptophan degradation to kynurenine) while the other is independent from its enzymatic function.

Potential clinical applications of kynurenine are growing. Kynurenine is required for the production of several metabolites, being the first metabolite in the kynurenine pathway, which leads to the final biosynthesis of nicotinamide adenine dinucleotide (NAD+). Among these metabolites, kynurenic acid and xanthurenic acid, as well as their analogues, have neuroprotective activities, suggesting that they may find use in drug development for neurological diseases [25,26,27,28]. Moreover, NAD+ is a coenzyme with a pivotal role in a vast number of biochemical processes, as well as in several redox reactions, and it is vital to mitochondrial function [29].

In this study, considering the only options for kynurenine production as chemical synthesis or extraction from human tissues, we expressed human IDO1 in plant cells in order to investigate the possibility to obtain an alternative production platform for both IDO1 protein and biologic kynurenine.

## 2. Results

### 2.1. Cloning of IDO1 cDNA in a Plant Expression Vector and Attempted Stably Expression in Tobacco Plants

A comparison of the human IDO1 amino acid sequence and the plant protein databases was performed. We found some similarity within plants and green algae only at a low percentage with an identity average of around 30%. Since alignment errors can appear when the sequence identity drops below 35% [30], these sequence homologies seemed to be too low to confirm the presence of IDO1 in the Plantae kingdom. Moreover, most of these proteins were reported to have an unknown function and, in the few cases of being identified as indoleamine 2,3-dioxygenase enzymes, the evidence supporting the existence was inferred only by homology.

Therefore, to express IDO1 in plants, we cloned the human IDO1 cDNA into plant expression vector pGreenII, obtaining pGreenII.IDO1. A schematic diagram of the vector assembly is shown in Figure 1A. Then, we transformed tobacco (*N. tabacum* cv. Petit Havana) plants with *A. tumefaciens* harbouring the pGreenII.IDO1 vector to obtain plants stably expressing human IDO1. Selection with kanamycin allowed for the regeneration of transformed plants (T0), as shown in Figure 1B (left panel). Regenerated shoots were successively separated from chlorotic leaf discs and placed in fresh selective medium to stimulate further growth and roots formation (Figure 1B, central panel). As expected, wild-type non-transformed plants failed to regenerate when placed in freshly selective medium (Figure 1B, right panel). Total DNA was isolated from leaves of selected regenerated T0 plants and examined by PCR (Figure 1C). Only transformed explants showed an amplified fragment of about 500 bp, confirming the presence of the IDO1 gene cassette in their genome. Few T0 events unable to amplify were considered as regeneration escapes. In total, 10 PCR-positive plants were chosen, transplanted to pots in controlled greenhouse conditions, and reared to flower for seed production. Seeds were germinated on kanamycin selective medium and for several plants (T1 events). IDO1 gene presence was confirmed by PCR analysis (Figure 1D). However, even if IDO1 mRNA was detected by RT-PCR in leaves (Figure 1E), IDO1 protein was undetectable by Western blotting in total proteins extracted from leaves and roots of transformed plants, suggesting a protein stability problem.

### 2.2. Human IDO1-GFP Can Be Transiently Expressed in Tobacco Protoplasts

The lack of human IDO1 expression in plants could possibly be attributed to protein instability. Therefore, a second cloning strategy was carried out by fusing IDO1 sequence to the GFP tag, aiming to obtain a more stable fusion protein, as frequently observed in fusion technology [31]. For this purpose and for obtaining a tag useful as both a protein detector and a possible purification system, we assembled the gene construct coding for IDO1-GFP under the same constitutive promoter previously used in pGreenII.IDO1, the 35S promoter (Figure 2A), and we transiently transfected tobacco protoplasts with the new vector pV.IDO1-GFP. By means of Western blotting with two different antibodies detecting IDO1 protein and the GFP tag, respectively, we revealed the expression of a 69 kDa protein, matching the size of IDO1-GFP, in plants transfected with pV.IDO1-GFP (Figure 2B). In order to avoid possibile impact of tryptophan depletion, due to the catalytic activity of IDO1, on cell vitality and protein expression, we added an extra amount of tryptophan to the protoplast medium. The additional tryptophan did not change IDO1 protein expression in protoplasts, as shown in (Figure 2B). As a result, IDO1 can be expressed in vitro as a GFP-fused protein in transiently transformed protoplasts.

### 2.3. Transiently Expressed IDO1-GFP Had Catalytic Activity and Similar Subcellular Localisation of Mammal IDO1

In mammal cells, IDO1 catalyzes the conversion of the essential amino acid tryptophan into kynurenine. Therefore, we investigated if IDO1-GFP transiently expressed in protoplasts had enzymatic activity. For this purpose, we measured kynurenine concentration in K3 medium after a 24 h incubation of protoplasts in medium alone or with extra tryptophan added to enhance and detect the metabolic product (Figure 3A). As a control, kynurenine was also measured in non-transformed protoplasts subjected to the same culture conditions. Human IDO1 expressed in protoplasts was found to be capable of producing kynurenine, but this effect was strictly related to the enzyme presence. In fact, the WT control protoplasts showed undetectable production of kynurenine in comparison with the IDO1-transfected cells, both in the presence and absence of an extra amount of tryptophan in the medium (Figure 3A). As an additional control, tobacco protoplasts were also transformed with the empty vector, and the kynurenine concentration measured in their K3 medium resulted in being undetectable, similar to what we found in WT protoplasts. Moreover, the addition of tryptophan to the medium allowed for the production of a significantly higher level of kynurenine as compared to untreated IDO1-transfected cells. Unexpectedly, the final (i.e., after the 24 h incubation) tryptophan level measured in the medium with extra-added tryptophan (100 µM) was significantly higher in the presence of transfected IDO1 cells than in the WT control (Figure 3B).

As recently demonstrated, in mammal cells, IDO1 is localised not only in the cytosol but also in early endosomes [24]. To investigate the intracellular localisation of human IDO1 in plant cells, we analyzed leaves of soil-grown tobacco plants transfected in vivo by *A. tumefaciens* cultures with both pV.IDO1-GFP and red fluorescent protein (RFP)-tagged organelle markers. Confocal analysis revealed that IDO1 was primarily found in a diffuse pattern consistent with cytoplasmic localisation. However, a partial compartmentalisation of IDO1 protein to prevacuolar compartments/multivesicular bodies (PVC/MVB) was also observed when co-expressed with a PVC/MVB marker (TR2RFP-VSR2) (Figure 3C). PVC/MVB are intermediate compartments mediating protein transport between trans-Golgi network (TGN) and vacuole, holding overlapping functions with mammalian early endosomes. Since plant organelles are not perinuclear as in mammalian cells, statistical analysis of co-localisation events could be performed to evaluate the relevance of the results. As seen in Figure 3C, Pearson’s correlation (rp) and Spearman correlation (rs) resulted in coefficients of 0.43 and 0.2, respectively. Since perfect co-localisation would result in statistical coefficients close to 1, our results confirmed the partial localisation of IDO1 at PVC/MVB membrane when co-expressed with TR2RFP-VSR2 (Figure 3C). On the contrary, IDO1 could not be found co-localising with other cell compartments of the secretory pathway when co-expressed with markers of the late prevacuolar compartment (RFP-Rha1, LPVC marker), Golgi apparatus (ST-RFP marker), or trans-Golgi network (RFP-SYP61, TGN marker) (Figure 3D).

As a whole, on the basis of these data, we found that GFP-tagged IDO1 can be transiently expressed in protoplasts in a catalytically active conformation and has a similar intracellular localisation to that found in mammalian cells.

## 3. Discussion

The production of biopharmaceuticals in plant has been named as “molecular farming” [32,33]. In particular, plants can be used to produce pharmaceutical proteins, such as replacement human proteins, antibodies, and vaccine candidates [3]. Several recombinant human proteins with therapeutic activity have been produced in plants, for example, the enzyme alpha-mannosidase is produced in stably transformed plants for the replacement therapy of the lysosomal storage disease alpha-mannosidosis. Another enzyme, named glucocerebrosidase, is produced in plant cell suspension cultures and commercialised [13] for Gaucher’s disease treatment. Furthermore, plant cell cultures represent interesting options for an easy and scalable production of secondary metabolites, such as the anticancer drug paclitaxel [34]. Thus, in addition to whole plants grown in greenhouse, plant in vitro systems for generating metabolites and recombinant proteins are now considered an attractive alternative to classical technologies [35,36]. Although the efficiency of plant platforms for biological components production is largely recognised, prokaryotic and mammalian expression systems are still mainly used in biotechnological processes because, provided with a strong regulatory framework, they guarantee robustness and economic sustainability for the industrial platform [3]. Nevertheless, plant platforms might be attractive for the production of certain biopharmaceuticals with favourable glycan configurations, as well as for a rapid scale up process in the production of recombinant protein. Moreover, plants certainly do not allow the growth of human pathogens [37,38].

In the current report, a cloning strategy was set for the expression of a catalytically active form of human IDO1 in plant cells, with the aim of exploring an alternative platform for the kynurenine production. This molecule is an agonist of AhR, a crucial factor in the immune response regulation that exerts its roles through different immunoregulatory mechanisms. In fact, when activated, AhR is capable of inhibiting the production of several proinflammatory cytokines and of controlling the generation of regulatory T (Treg) cells and/or T helper 17 (Th17) cells in mouse models of autoimmune diseases [39]. The finding of possible kynurenine participation in a positive feedback loop in AhR signalling paves the way to a probable use of kynurenine as an immunoregulatory drug.

Plant organisms naturally produce all amino acids including tryptophan, which not only result as an essential component of protein synthesis, but also serves as a precursor for a wide range of indole-containing secondary metabolites that are important for plant growth, such as auxin [40]. Therefore, the catalytic activity of recombinant human IDO1 in transformed plants could be exploited to produce kynurenine in an organism normally producing large quantities of its precursor. Apart from arylformamidase, plants do not possess orthologous genes coding for enzymes involved in the kynurenine pathway [41]. Rice (Oryza sativa) seems to be an exception because genomic database searching has revealed the presence of several biosynthetic genes encoding core enzymes in kynurenine pathways, despite a lack of biochemical evidence [41]. In particular, the sequence of an IDO1 gene (OsIDO) was identified, coding for a protein with a low identity (26%) and low similarity (42%) to *Mus musculus* IDO1. However, the enzymatic activity of OsIDO had not been demonstrated [42]. Hence, during our study, we generated a specific vector for efficient transfection of human IDO1 gene in plants. Although the genetic transformation had been effective and the transgene transcription did occur, transformed plants failed to express IDO1 protein. A possible explanation could be the instability and/or the rapid degradation of human IDO1 protein in a heterologous platform such as the plant cell. Therefore, a different construct was cloned encoding for the fusion protein IDO1-GFP with the aim of improving the protein stability. In order to rapidly test the expression of this new gene construct, we transiently transfected protoplasts, plant cells with their cell wall removed, and cultivated them in vitro in a liquid medium. Under these conditions, IDO1-GFP fusion protein was expressed, accumulated in the protoplasts and, most importantly, secreted kynurenine in the medium. Heterologous protein expression and kynurenine production in protoplasts were also assessed in the presence of an excess of tryptophan in order to compensate the depletion caused by the IDO1-metabolising activity that could interfere with the plant cell development, as recently demonstrated. In fact, according to He and colleagues [43], kynurenine is an alternative substrate that can competitively inhibit the key enzymes involved in the conversion of tryptophan into indole-3-pyruvic acid in one of the auxin biosynthesis pathways. They indicated the structural similarity between kynurenine and tryptophan as the cause of kynurenine-mediated inhibition of auxin biosynthesis in roots of *Arabidopsis thaliana* seedlings when grown in a medium containing kynurenine. Interestingly, they found that the effect of kynurenine treatment could be suppressed by the presence in the medium of high doses of tryptophan. In our study, we observed that the extra tryptophan supplementation to the protoplast medium, rather than affecting the catalytic activity of IDO1-GFP, increased fourfold the production of kynurenine. These findings suggest that kynurenine production increases according to the availability of the substrate tryptophan in the medium, and IDO1 activity promotes an increased level of tryptophan in the medium by a still unidentified mechanism. The latter aspect is not less important for a nutraceutical application. In fact, tryptophan for humans is an essential amino acid necessary for the in vivo biosynthesis of proteins, as well as a precursor of several bioactive compounds. It is widely used as a dietary supplement for several perceived benefits, such as sleep and mood regulation [44], and it is reported to mitigate the course of several chronic diseases, such as Chron’s disease [45]. Finally, in plants, IDO1 keeps the same dual localisation recently found in mammalian cells [46]. In fact, when expressed in vivo in tobacco leaves via *Agrobacterium* infiltration and observed by confocal microscopy, the majority of IDO1-GFP are found in the cytoplasm, with partial colocalisation with the early endosomes.

In summary, this study demonstrates that in plant cells IDO1-GFP shares the same enzymatic activity and subcellular distribution with native IDO1, paving the way to the possible use of plants as bioreactors to produce kynurenine. To the best of our knowledge, this is the first report on kynurenine production in plant cells by expressing human IDO1 protein. Further studies will be necessary to verify if a negative IDO1-mediated interference with plant metabolism may occur during plant growth.

## 4. Materials and Methods

### 4.1. Construction of Plasmids for Plant Transformation

To stably express IDO1 in tobacco plants, we excised human IDO1 cDNA from plasmid pEZM02.IDO1 using *Eco*RI and *Not*I restriction enzymes. After excision, both 5′ and 3′ sticky ends were blunted by DNA polymerase I large (Klenow) fragment to allow non-compatible ends joining. The result of digestion was separated on a 1% agarose gel and the IDO1 band was isolated and purified using the T-Pro Gel/PCR DNA Purification kit (T-Pro Biotechnology, New Taipei County, Taiwan). The plant vector pDHA [47] was linearised with *Bam*HI restriction enzyme, blunted as described above, dephosphorylated using calf intestinal alkaline phosphatase (CIAP), and ligated to the IDO1 fragment to obtain the pDHA.IDO1 intermediate vector, where IDO1 was under the control of the constitutive 35S cauliflower mosaic virus (CMV) promoter and the 35S terminator suitable for plant expression. For the production of transgenic plants, we introduced the expression cassette excised by *Eco*RI digestion from the pDHA.IDO1 plasmid into the pGreenII.0029 binary vector [7] opened with the same restriction enzyme, obtaining pGreen.IDO1. This final plasmid was sequenced to verify the correct sequence and orientation of the IDO1 expression cassette. Strain GV3101 of *A. tumefaciens* was transformed by electroporation with the pGreenII.IDO1 vector and used to produce transgenic tobacco (*N. tabacum* cv. Petit Havana), as described by Pompa and colleagues [48]. Briefly, tobacco plants were transformed by co-cultivation with *A. tumefaciens* harbouring pGreenII.IDO1 on agar-solidified MS medium containing 1 mg/L of 6-benzylaminopurine (BA) and 0.1 mg/L of α-naphthaleneacetic acid (NAA) hormones. After co-cultivation, the leaf discs were grown aseptically in Petri dishes at 25 °C under full light on the same medium supplemented with 500 mg/L of cefotaxime and 100 mg/L kanamycin to avoid *Agrobacterium* growth and to select transformed cells, respectively. Shoots emerging from each explant were kept separated to guarantee regeneration of independent transformants. After approximately 5–6 weeks, shoots were plated on MS medium without hormones but still supplemented with antibiotics until new plants developed. Transformed plants (T0) were grown at 25 °C in 16 h of light in axenic culture in glass jars without antibiotics and propagated every 5–6 weeks. After PCR results, 10 transgenic plants (labelled with numbers from 1 to 10) were chosen and reared to flower to obtain seeds. T1 seeds from self-fertilisation were harvested at maturity, seeded onto kanamycin selective medium, and scored for resistance. After 21–28 days, resistant seeds started to germinate. One T1 plant deriving from each of the 10 primary regenerants, T1n (*n* = 1–10), was analysed by PCR to confirm the IDO1 gene presence. To fuse GFP in frame to IDO1 at the C-terminal, we amplified IDO1 cDNA with PCR from pDHA.IDO1 with the forward primer IDO1ClaI (GCAGACTACATCGATGGCACACGCTATGG) and the reverse primer IDO1NheI (GGCCGCTAGCACCTTCCTTCAAAAG). The PCR fragment was digested by *Cla*I and *Nhe*I restriction enzymes and inserted in the vector pJA1, derived from pAW7 after replacing the RFP with GFP [49]. The resulting vector (pUC.IDO1-GFP) contained the 35S promoter and the IDO1 sequence in frame with a C-terminal GFP, followed by the NOS terminator in a *Eco*RI-*Hind*III cassette. Since IDO1 coding sequence contained another *Hind*III site (Figure 2A), the digestion of pUC.IDO1-GFP by *Eco*RI/*Hind*III generated two fragments. Both fragments were cloned in the pV binary vector for plant transformation and digested with *Eco*RI/*Hind*III in order to reconstitute the original IDO1-GFP. This reconstituted cassette was sequenced to confirm the integrity of IDO1 gene and the frame with GFP. The resulting vector pV.IDO1-GFP was used for tobacco protoplast transformation and, after transformation into *A. tumefaciens*, for infiltration experiments.

### 4.2. Tobacco Protoplast Isolation

Protoplasts were isolated from transgenic plants as described by De Marchis et al. [50]. Briefly, to isolate mesophyll protoplasts, we wounded young fully expanded leaves from tobacco plants grown under aseptic conditions with a scalpel and placed them in a Petri dish with the abaxial side in contact with the 1× enzyme mix solution (in 1% Cellulase Onozuka RS, 0.5% Macerozime R-10) dissolved in K3 medium (3.78 g Gamborg’s B-5 Basal Medium with Minimal Organics, 5.1 mM CaCl_2_·2H_2_O, 3.12 mM NH_4_NO_3_, 0.4 M sucrose, 1.67 mM xylose, BA 1 mg/L, NAA 1 mg/L (pH 5.5)) for 16 h at 25 °C in the dark. Protoplasts were recovered by resuspending macerated leaves in K3 medium, filtering through a sterile 85 μm nylon mesh, and centrifuging for 20 min at 60× *g* with a swinging bucket rotor. The floating green band, representing vital protoplast, was recovered and washed twice with W5 solution (152 mM NaCl, 5 mM KCl, 125 mM CaCl_2_·2H_2_O, 5 mM glucose). After testing viability and yield with fluorescein diacetate staining, we used protoplasts for further analysis.

### 4.3. Tobacco Protoplast Transformation

After W5 removal by centrifuging for 10 min at 60× *g*, protoplasts were resuspended at a concentration of 1 × 10^6^ mL^−1^ with MaCa buffer (0.1% MES, 20 mM CaCl_2_, 0.5 M mannitol (pH 5.6)), heated at 45 °C for 5 min, and mixed with 60 µg of plasmid DNA mL^−1^ of protoplast suspension. Protoplast transformation with polyethylene glycol (PEG) was performed by adding an equal volume of a solution containing 40% (*w*/*v*) PEG 4000, 0.1 M Ca(NO_3_)_2_, and 0.4 M mannitol (pH 8). After incubation at room temperature for 30 min, transformed protoplasts were slowly diluted to 14 mL with W5 solution. Protoplasts were pelleted at 60× *g* for 10 min at room temperature, resuspended in K3 medium to a final density of 1 × 10^6^ protoplasts mL^−1^, and incubated 2 h at 25 °C in the dark. Then, tryptophan 100 µM was added to half of the samples, and all the protoplasts were incubated for 24 h at 25 °C in the dark before subsequent analysis.

### 4.4. Plant Transformation by Agroinfiltration and Microscopy Analysis

Soil-grown *N. tabacum* cv. Petit Havana [51] leaves were infiltrated at OD 0.1 with *A. tumefaciens* cultures containing the pV-IDO1-GFP alone or with either organelle marker such as ST-RFP (Golgi marker), RFP-SYP61 (TGN marker), TR2RFP-VSR2 (PVC/MVB marker), or RFP-Rha1 (LPVC marker), as described previously [52]. The infiltrated areas were analysed 48h post-infiltration by confocal laser scanning microscopy. Confocal imaging was performed using an inverted Zeiss LSM 700 laser scanning microscope. When GFP fusions were imaged with RFP organelle markers, samples were excited with a diode laser at the wavelength of 488 nm for GFP and 555 nm for RFP. Fluorescence was detected with a 552 nm dichroic beam splitter and a 475 to 550 nm bandpass filter for GFP and a 560 to 700 nm bandpass filter for RFP. All dual-colour imaging was performed by line switching to obtain adequate live bioimaging data that had not been distorted by organelle motion. Statistical colocalisation between IDO1-GFP and TR2RFP-VSR2 was performed as described in Gershlick et al. [53].

### 4.5. DNA Isolation and Analysis

Total DNA was isolated from wild type and transgenic tobacco leaves using the GenElute™ Plant Genomic DNA Miniprep kit (Merck KGaA, Darmstadt, Germany). DNA PCR amplification was carried out using the primer pairs forward 35S1 (5′-ACGTTCCAACCACGTCTTCAAAG-3′) and reverse IDO1 (5′-GCAAGACCTTACGGACATCT-3′) and forward 35S2 (5′-CATGGAGTCAAAGATTCAAA-3′) and reverse IDO1 for T0 and T1 plants, respectively.

### 4.6. RNA Extraction and RT-PCR Analysis

Total RNA was extracted from leaf tissues of transformed plants with the NucleoSpin^®^ RNA Plant kit (Macherey–Nagel, Düren, Germany). For the reverse-transcribed PCR (RT-PCR) analysis, RNA was treated with DNase I (Qiagen, Venlo, The Netherlands) and reverse-transcribed (2 µg) using oligo(dT)18 and the SuperScript III Reverse Transcriptase (Thermo Fisher Scientific, Waltham, MA, USA), according to the manufacturer’s instructions. The cDNA preparations were analysed by PCR using specific primers for human IDO1 (forward, 5′- CTCCTGGACAATCAGTAAAGAGTACC-3′; reverse, 5′- ACTTCTCAACTCTTTCTCGAAGCTGG-3′) and plant actin (forward, 5′- ATGGCGGATGGGGAGGACATTCA-3′; reverse, 5′- CCTTTTGTTATCCACATCTGTTG-3′).

### 4.7. Protein Extraction from Protoplasts and Western Blotting Analysis

The protoplasts were washed twice in cold PBS and lysed with Laemmli sample buffer. Proteins were separated by SDS-PAGE on 10% polyacrylamide gels and then transferred to nitrocellulose. Membranes were blocked with 5% (*w*/*v*) non-fat dried milk in Tris-buffered saline containing 0.1% (*v*/*v*) Tween 20 at room temperature for 1 h. After being blocked, membranes were incubated overnight with primary antibodies, namely, anti-IDO1 (Merck Millipore, Darmstadt, Germany), anti-GFP (Thermo Fisher Scientific, Waltham, MA, USA), and anti-Rubisco large subunit (rbcL, Agrisera, Sweden), then washed and incubated with appropriate horseradish peroxidase (HRP)-conjugated secondary antibody (Thermo Fisher Scientific, Waltham, MA, USA) for 1 h at room temperature. Immunoreactive bands were detected by enhanced chemiluminescence (Bio-Rad, Hercules, CA, USA).

### 4.8. Kynurenine Analysis

IDO1 functional activity was measured in protoplast culture supernatants in terms of the ability to metabolise tryptophan to kynurenine, whose concentrations were measured by using a Perkin Elmer, series 200 HPLC instrument (Waltham, MA, USA). A Kinetex^®^ C18 column (250 × 4.6 µm, 5 μm, 100 Å; Phenomenex, USA), maintained at the temperature of 37 °C and pressure of 1800 PSI, was used. Briefly, 1 × 10^6^ transformed protoplasts were resuspended in 1 mL of K3 medium alone or supplemented with exogenous L-tryptophan 100 µM (Sigma-Aldrich, St. Louis, MO, USA) which was added 2 h after transfection. Then, following an incubation of 24 h at 25 °C in the dark, the K3 medium was recovered with a Pasteur pipette and filtered with a syringe containing a cotton filter to eliminate remaining protoplasts. After deproteinisation, the samples were eluted by a mobile phase containing 10 mM NaH_2_PO_4_ (pH = 3.0; 99%) and methanol (1%) (Sigma-Aldrich, St. Louis, MO, USA), with a flow rate of 1.3 mL/min. A UV detector at 360 nm and 220 nm performed detection of kynurenine and tryptophan, respectively. The software TURBOCHROM 4 was used for evaluating the concentrations of kynurenine and tryptophan in the samples by mean of a calibration curve.

### 4.9. Statistics

Data are representative of at least three independent experiments. All results are shown as mean ± SD. One-way ANOVA followed by post hoc Bonferroni’s test was used when three or more samples were under comparison. All analyses were performed using Graph Pad Prism software 8.0.

## Figures and Tables

**Figure 1 ijms-22-05102-f001:**
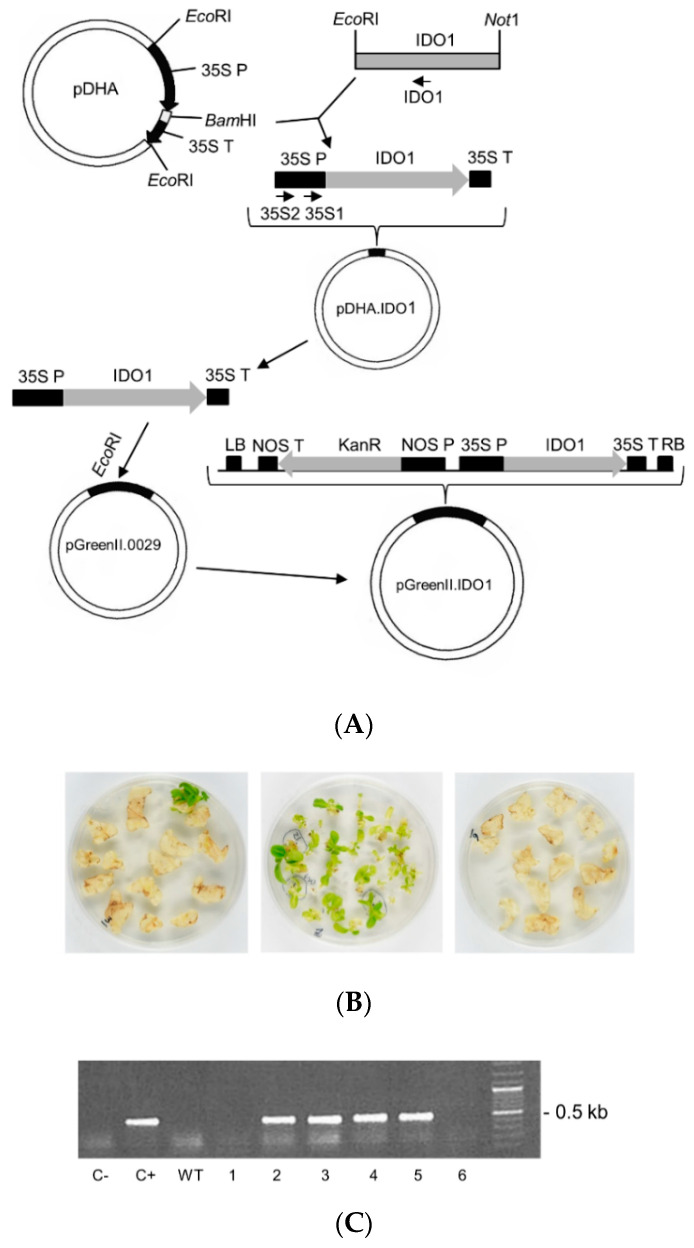
Attempt of human IDO1 protein stable expression in tobacco plants. (**A**) Schematic diagram of pGreen.IDO1 vector cloning. Abbreviations: 35S P, 35S promoter sequence; 35S T, 35S terminator sequence; NOS P and NOS T, *A. tumefaciens* nopaline synthase (*nos)* promoter and terminator sequence, respectively; LB, left border; RB, right border; KanR, gene encoding for kanamycin resistance. The annealing positions of primers 35S1, 35S2, and IDO1 are shown. (**B**) Transgenic tobacco plant selection and regeneration on MS medium with kanamycin after *A. tumefaciens* co-cultivation. Only transformed cells were able to regenerate on selective medium (left panel). Regenerated shoots were separated from chlorotic leaf discs and sub-cultured to allow growth and rooting (central panel). No regeneration occurred when WT leaf discs were placed on antibiotic selective medium (right panel). (**C**) PCR on some tobacco primary regeneration events (T0) with primers 35S1/IDO1. The amplification signal was detectable only in transformed plants (2–5), whereas plants 1 and 6 were probably regeneration escapes. (**D**) One T1 plant deriving from seeds of each of the 10 primary transformants (T1_n; n = 1–10_) were analysed by PCR with primers 35S2/IDO1, confirming the presence of the IDO1 transgene. There were two empty lanes between the lanes WT and T1_1_. (**E**) Different transformed plants (2−5) were analysed for the presence of IDO1 mRNA by RT-PCR. Actin was amplified as a reference gene for the RT-PCR experiment. (**C**–**E**) WT: wild type plant; (**C**,**D**) C+: pGreen.IDO1 vector; C-: no DNA.

**Figure 2 ijms-22-05102-f002:**
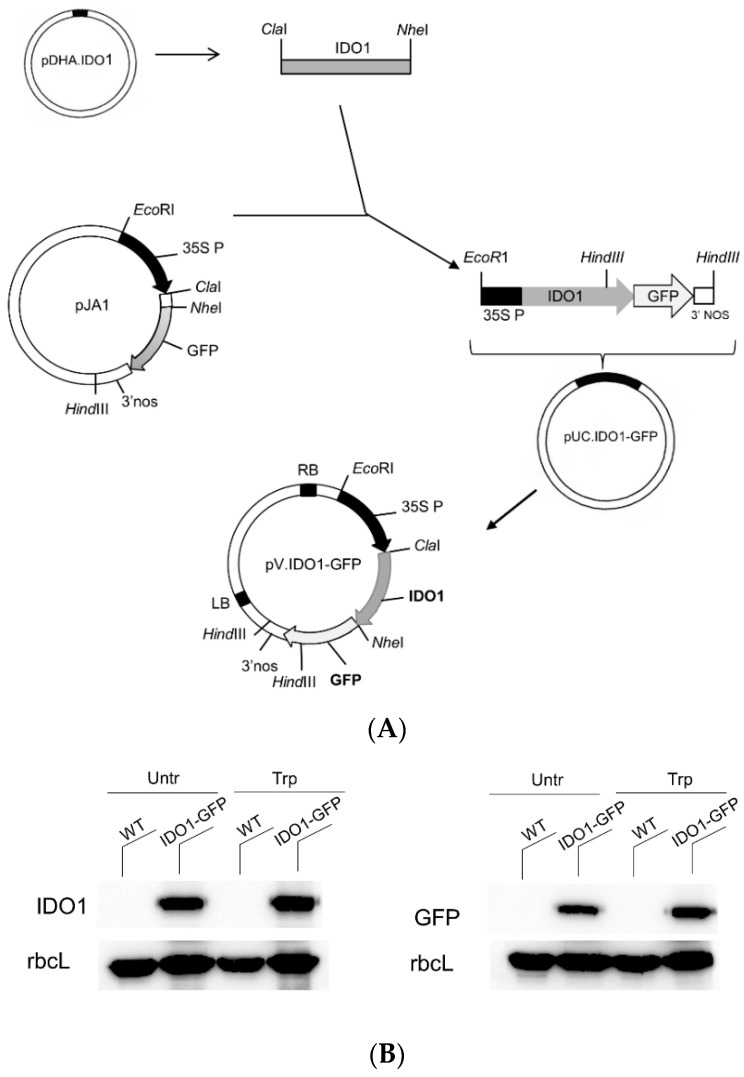
IDO1-GFP expression in protoplasts. (**A**) Schematic diagram of pV.IDO1-GFP construct cloning. Abbreviations are the same of Figure 1A. (**B**) Western blot analysis. IDO1 accumulation was analysed in whole-cell lysates of protoplasts transfected (IDO1) or not (WT, wild type) with pV.IDO1-GFP vector by using both an IDO1 and a GFP specific antibody. An antibody against rubisco large subunit (rbcL) was used to verify protein normalisation. After transformation, half of the protoplasts were incubated with tryptophan 100 µM (Trp), and the other half remained untreated (Untr). One representative immunoblot analysis of three is shown.

**Figure 3 ijms-22-05102-f003:**
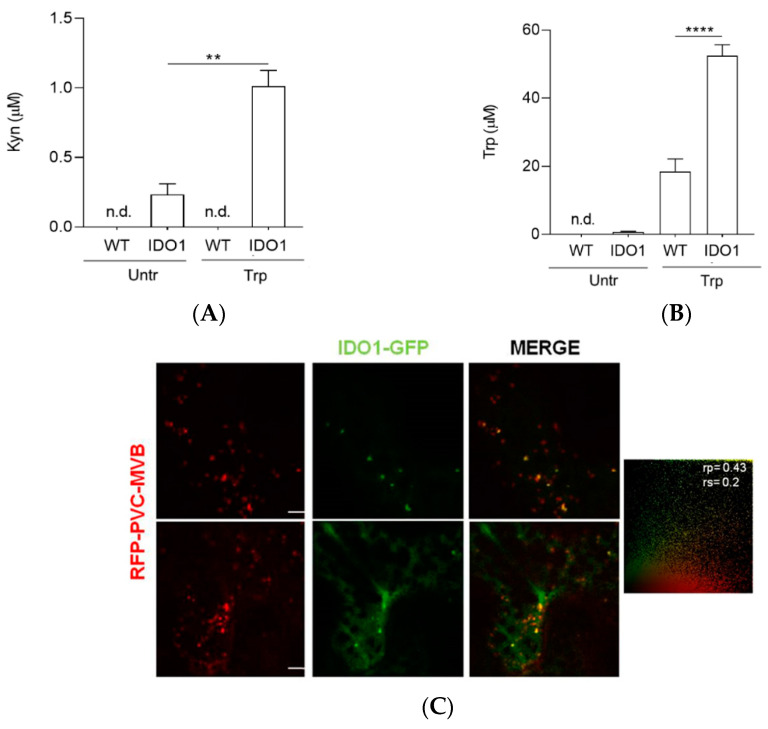
(**A**) Kynurenine (Kyn) and (**B**) tryptophan (Trp) concentrations were assessed in the medium of transformed (IDO1) or untransformed (WT, wild type) protoplasts maintained for 24 h in culture in the presence (Trp) or absence (Untr) of tryptophan excess (100 µM). Concentrations below the detection limit are indicated as not detectable (n.d.). In (**A**,**B**), data are means ± S.D. of three experiments, each performed in triplicate. Data were analysed by one-way ANOVA followed by Bonferroni’s post hoc test. ** *p* < 0.01; **** *p* < 0.0001. (**C**,**D**) Confocal immunofluorescence microscopy images of *Agrobacterium*-infiltrated tobacco leaf cells coexpressing IDO1-GFP with various organelle markers carrying an RFP tag. The cells are shown in three-channel mode (green, red, and merge). Scale bar, 5 nm. (**C**) A partial compartmentalisation of IDO1 protein (green channel) to the prevacuolar compartments/multivesicular bodies (PVC/MVB) marker (red channel) was observed in the merged image. The statistical colocalisation between IDO1 and PVC/MVB marker is shown, with Pearson’s correlation coefficient (rp) of 0.43 and Spearman correlation coefficient (rs) of 0.2. (**D**) IDO1 did not seem to colocalise with cell compartments of the secretory pathway such as the Golgi, the trans-Golgi network (TGN), or the late prevacuolar compartment (LPVC).

## Data Availability

Not applicable.

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
