# Peer review of "Human Indoleamine 2,3-dioxygenase 1 (IDO1) Expressed in Plant Cells Induces Kynurenine Production"

_ijms, 2021, doi:10.3390/ijms22105102_

Round 1
Reviewer 1 Report
In this study, Bellucci, Pompa et al. described experiments performed to ensure the expression of a gene encoding human indoleamine 2,3-dioxygenase 1 (IDO1) in plants (Nicotiana tabacum) for production of tryptophan degradation product - kynurenine. They used classical transformation by Agrobacterium tumefaciens. Unfortunately, no protein product of the transformed gene could be detected. Therefore, they proceeded to construct a vector for transient expression of sequences encoding the IDO1- GFP fusion protein and after transformation of protoplasts with this vector, they detected both the gene product and the production of kynurenine. In this study, the authors did not complete their intention to create transgenic kynurenine-producing plants, nevertheless, they showed a possible way.
Comments:
The construction of vectors, the process of transformation and verification of the presence of cloned DNA sequences (by PCR method but not by sequencing) and the protein product (by western blotting) represent a substantial part of the results. In the first cloning experiment, the authors reported that mRNA for IDO1 was synthesized (results, line 113 and discussion, line 266) in various parts of the plants. However, the data are not shown, although this result is essential for the author´s conclusion that the undetected protein product is unstable or rapidly degraded in plants.
The concentration of kynurenine in the medium of transiently transformed protoplasts was approx. 1uM, but it would be good to give the yield of kynurenine production per number of transfected cells.
The text should be checked for minor grammatical errors and punctuation.
Author Response
Point 1. The construction of vectors, the process of transformation and verification of the presence of cloned DNA sequences (by PCR method but not by sequencing) and the protein product (by western blotting) represent a substantial part of the results. In the first cloning experiment, the authors reported that mRNAfor IDO1 was synthesized (results, line 113 and discussion, line266) in various parts of the plants. However, the data are not shown, although this result is essential for the author´s conclusion that the undetected protein product is unstable or rapidly degraded in plants.
Response 1: We agree with the Reviewer that the demonstration that IDO1 mRNA is synthesized in transformed plants is essential for our conclusions. As concerns the mRNA synthesized in various parts of the plants, there is a misunderstanding because we reported in the first version of the manuscript that proteins were extracted from different tissues for the western blotting, not for mRNA detection. Therefore, to clarify this point and show the results, in the revised paper we have added IDO1 RT-PCR as Figure 1e, with the corresponding part in the Materials and Methods. The text in the results is now: “However, even if IDO1 mRNA was detected by RT-PCR in leaves (Figure 1e), IDO1 protein was undetectable by western blotting in total proteins extracted from leaves and roots of transformed plants, suggesting a protein stability problem.”
Point 2. The concentration of kynurenine in the medium of transiently transformed protoplasts was approx. 1uM, but it would be good to give the yield of kynurenine production per number of transfected cells.
Response 2: The Reviewer’s observation is well done. Kynurenine concentrations are referred to the supernatants harvested from 1 x 106 of transformed protoplasts suspended in 1 ml of medium alone or supplemented with exogenous tryptophan for 24h at 25°C. Published data described that the transformation efficiency of tobacco protoplasts is between 40%-60% (Efficiency of transient transformation in tobacco protoplasts is independent of plasmid amount. Locatelli F, Vannini C, Magnani E, Coraggio I, Bracale M. Plant Cell Rep. 2003 Jun;21(9):865-71. doi: 10.1007/s00299-003-0593-x. Epub 2003 Mar 7).
Therefore, taking into account a similar transformation efficiency, we can estimate that the yield of kynurenine production in our setting could be around 0.125 µmol/l and 0.5 µmol/l per 1 x 106 transformed protoplasts. Therefore, in order to address the main concern of the Reviewer and to provide readers with more detailed information about the kynurenine production from IDO1 transformed protoplasts, we have now modified Materials and Methods.
Point 3. The text should be checked for minor grammatical errors and punctuation.
Response 3: We apologize for the occurrence of grammatical errors and punctuation. The new version of the manuscript has been carefully revised for the English language.

Reviewer 2 Report
Dear Author,
The manuscript has only minor corrections as incorporated in PDF attached copy.

Author Response
REVIEWER #2
Point 1. Attached pdf file.
Response 1: The formatting errors described in the attached pdf file have been rectified as suggested by the Reviewer. We could not add more keywords because the maximum number of keywords allowed by the journal is three. Moreover, the sentence at line 310 “was excised from plasmid pEZM02.IDO1 using…” was not modified because we do not understand which is the Reviewer suggestion in this specific case.
Reviewer 3 Report
In the manuscript entitled “Human indoleamine 2,3-dioxygenase 1 (IDO1) expressed in 2 plant cells induces kynurenine production” authors describe a procedure, the aim of which was to obtain a transgenic plant capable of production indoleamine 2,3-dioxygenase 1 (IDO1) to obtain kynurenine as a result. Since the main option for kynurenine production are chemical synthesis or extraction from human tissues, authors tried to expressed human IDO1 in plant cells.
Questions:
- Authors confirmed the presence of an appropriate insert in the construct only by PCR reaction with specific primers. It should rather be sequenced to confirm the proper connection, orientation and sequence, especially when using such a cloning tactic - for blunt ends. Sequencing is mentioned only in the description of the step resulting the construct for the protoplast transfection and agroinfiltration.
- It is a pity that the authors did not show photos of the transformed protoplasts, it could confirm the potential location.
- 3 c and d - the description of this illustrations is not sufficient
- Lack of the italic in many places
Author Response
REVIEWER #3
Point 1: Authors confirmed the presence of an appropriate insert in the construct only by PCR reaction with specific primers. It should rather be sequenced to confirm the proper connection, orientation and sequence, especially when using such a cloning tactic - for blunt ends. Sequencing is mentioned only in the description of the step resulting the construct for the protoplast transfection and agroinfiltration.
Response 1: The point raised by the Reviewer is extremely important because after cloning of genes in final plasmid vectors is mandatory to check that the inserted DNA sequence is not mutated and that has a correct orientation. We sequenced the IDO1 expression cassettes in both pGreen.IDO1 and pV.IDO1-GFP vectors, but we forgot to insert the pGreen.IDO1 sequencing in the Materials and Methods and we are sorry for this. Therefore, now a sentence reporting this detail has been included in the revised Materials and Methods of the manuscript.
Point 2: It is a pity that the authors did not show photos of the transformed protoplasts, it could confirm the potential location.
Response 2: We agree with the reviewer that confocal images of IDO1-GFP transformed protoplasts could be useful to confirm the enzyme intracellular localization. Unfortunately, we performed confocal experiments of transiently transformed protoplasts, but we did not obtain good quality images due to poor GFP fluorescence. Therefore, for this reason, we decided to use transiently transfected plant tissues and we indeed obtained better results.
Point 3: 3 c and d - the description of this illustrations is not sufficient.
Response 3: The description of illustrations in Figure 3c and 3d have been improved, both in the text and in the figure legend.
Point 4: Lack of the italic in many places.
Response 4: We apologize for the lacking of italic and formatting errors, which have been rectified.
Round 2
Reviewer 1 Report
I have no other objections (except the small correction in the added text :line 439 1 x 106 should be 1 x 106